# Effects of Calorie Restriction on Health Span and Insulin Resistance: Classic Calorie Restriction Diet vs. Ketosis-Inducing Diet

**DOI:** 10.3390/nu13041302

**Published:** 2021-04-15

**Authors:** Ana Napoleão, Lívia Fernandes, Cátia Miranda, Ana Paula Marum

**Affiliations:** 1Spanish Society of Precision Health, SENMO/SESAP, 38320 Canary Islands, Spain; anacnapoleao@gmail.com; 2Integrative Medicine Centre, CEMINT, Technical-Scientific Nucleus, 2800-56 Almada, Portugal; livia.h.fernandes@hotmail.com (L.F.); katia.nutricm@gmail.com (C.M.); 3Polytechnic Institute of Setúbal, Escola Superior de Saúde do Instituto Politécnico de Setúbal, 2910-761 Setúbal, Portugal

**Keywords:** diet, calorie restriction, ketosis, fasting, health span, lifespan, metabolic syndrome, insulin resistance, chronic non-communicable diseases, low-calorie, low-carb

## Abstract

As the incidence of Chronic Non-Communicable Diseases (CNCDs) increases, preventive approaches become more crucial. In this review, calorie restriction (CR) effects on human beings were evaluated, comparing the benefits and risks of different CR diets: classic CR vs. ketosis-inducing diets, including intermittent fasting (IF), classic ketogenic diet (CKD), fasting mimicking diet (FMD), very-low-calorie ketogenic Diet (VLCKD) and Spanish ketogenic Mediterranean diet (SKMD). Special emphasis on insulin resistance (IR) was placed, as it mediates metabolic syndrome (MS), a known risk factor for CNCD, and is predictive of MS diagnosis. CR is the most robust intervention known to increase lifespan and health span, with high evidence and known biochemical mechanisms. CR improves cardiometabolic risk parameters, boosts exercise insulin sensitivity response, and there may be benefits of implementing moderate CR on healthy young and middle-aged individuals. However, there is insufficient evidence to support long-term CR. CKD is effective for weight and MS management, and may have additional benefits such as prevention of muscle loss and appetite control. SKMD has extreme significance benefits for all the metabolic parameters studied. Studies show inconsistent benefits of IF compared to classic CR. More studies are required to study biochemical parameters, reinforce evidence, identify risks, and seek effective and safe nutritional CR approaches.

## 1. Introduction

In recent years, disease prevention, preservation of good health, health span and life span have been regaining a special interest of global society and the scientific/medical community’s attention. At the same time, we have been witnessing a marked increase in human wellbeing and longevity, together with an increase in the prevalence of chronic diseases and medical costs. Taken together, these issues raise crucial questions on how we can improve health from a young age, and consequently, improve the quality of life, healthy aging and lifespan. It is known that adequate nutrition is correlated with good health [1,2]. Therefore, it is crucial to understand how different diets can affect health and lifespan, and how can different dietary approaches reduce the risk of developing chronic non-communicable diseases (CNCDs). To date, evidence has shown that a reduced caloric intake, without malnutrition, represents the most robust intervention to increase life span in model organisms, including primates, and to delay the emergence of age-related diseases [1]. Nevertheless, other dietary approaches, such as fasting and ketogenic diets, are emerging with compelling adherence, and therefore the need to understand their various impacts on health span and lifespan.

This review discusses the effects of calorie restriction (CR) through different diets—classic calorie restriction vs. calorie restriction with ketosis-inducing diets—in insulin levels, insulin resistance (IR) and glucose metabolism, associated with metabolic health and health span. The focus will be on healthy non-obese individuals.

For the research of bibliography on PubMed and SciELO databases, a combination of the following medical subject headings (MeSH) was used: calorie restriction, fasting mimicking diets, fasting, ketogenic diet, metabolic syndrome, insulin resistance, healthy aging, performance.

The reasons behind the choice of “insulin resistance” as a health span indicator are the well-established roles of IR as a powerful risk factor for CNCD—Type 2 Diabetes Mellitus (T2DM), coronary inflammation, heart disease, some forms of cancer, and others. IR predicts the risk of developing T2DM as early as thirty years before diagnosis. Moreover, IR and the compensatory hyperinsulinemia play a role in the pathogenesis of hypertension, inhibition of fibrinolysis, and stimulation of vascular smooth muscle proliferation and migration, all leading to atherosclerosis [3]. Insulin also plays a pivotal role in other common diseases that affect the quality of life of individuals, such as acne and polycystic ovary syndrome [4].

The acronyms used throughout the text are listed in the attached Table A1.

## 2. Calorie Restriction, Definition and Effects

Calorie restriction (CR) is a nutritional intervention of reduced energy intake of about 25–30% without lack of essential nutrients [1], with a well-established role to extend health span and lifespan in rodent and primate models. Throughout the literature, the term CR is often used interchangeably with dietary restriction (DR). However, CR is a partial example of DR as DR protocols include CR, reduction of specific macronutrients or change in the ratio of them as well as other different feeding interventions, such as ketosis-inducing diets [1].

Observational and randomized clinical trials denote that some of the mechanisms that lead to these improvements in animal models can also be seen in humans [5,6]. In particular, moderate CR in humans enhances multiple metabolic and hormonal factors that are implicated in the pathogenesis of age-associated metabolic alterations, cancer and others, which are leading causes of morbidity, disability, and mortality. Furthermore, moderate CR can prevent and reverse the harmful effects of the accumulation of excessive body fat and obesity, T2DM, dyslipidemia, hypertension, all factors included in metabolic syndrome (MS) [5,6]. IR is one of the most significant factors implicated in these results.

### 2.1. Calorie Restriction, Effects on Lifespan and Health Span

Healthy aging refers to the delay of molecular and cellular decline for the longest length of the lifespan, as aging is characterized by the accumulation of molecular and cellular damage, leading to structural and functional aberrancies in cells and tissues [7]. CR is the most robust intervention known to increase maximal lifespan and health span. The four most important CR-induced antiaging mechanisms are thought to be: neuroendocrine system adaptations, prevention of inflammation, hormetic response, and protection against oxidative stress damage [5]. Regarding neuroendocrine system adaptations, important involved factors are increased insulin sensitivity, reduced levels of anabolic hormones (e.g., insulin, testosterone, leptin), reduced levels of hormones that regulate thermogenesis and cellular metabolism (e.g., triiodothyronine, norepinephrine), and increased levels of hormones that suppress inflammation (e.g., cortisol, adiponectin, ghrelin) [5]. In rodents, CR prevents or delays a wide range of CNCD, such as cancer, atherosclerosis, diabetes, cardiomyopathy, kidney disease, autoimmune and neurodegenerative diseases [5].

Hormesis is another proposed factor to mediate the antiaging effects of chronic CR. Hormesis is defined as a beneficial biological process by which a low-intensity stressor increases resistance to another more intense stressor, by provoking a survival response in the organism, helping it to endure adversity by activating longevity pathways [5]. This theory can explain why CR animals are more resistant to a wide range of stresses (e.g., surgery, radiation, acute inflammation, exposure to heat, and oxidative stress) [5].

CR has also been shown to enhance DNA repair systems, promote the removal of damaged proteins and oxidized lipids, and upregulate endogenous enzymatic and nonenzymatic antioxidative defense mechanisms [5]. On the cellular level, CR triggers processes such as activation of cellular stress response elements, improved autophagy and modification of apoptosis [8]. CR reduces fasting insulin levels, several growth factors, profibrotic molecules, and cytokines, including serum concentration of platelet derived growth-factor (PDGF), transforming growth factor α (TGF-α), and tumor necrosis factor α (TNF-α) [5].

Biological molecular pathways that have the dual ability to sense nutrients and/or energy levels while also regulating cellular processes like epigenomic remodeling, gene expression, protein activity, and organelle integrity—i.e., mechanistic target of rapamycin (mTOR), insulin/insulin-like growth factor 1 signaling (ISS), AMP-activated protein Kinase (AMPK), sirtuins (SIRT)—each play a key role in aging [7]. Alongside, spontaneous or induced genetic alterations on genes encoding proteins of the IIS pathway can extend maximal lifespan in animal models, and that CR decreases serum insulin-like growth factor 1 (IGF-1) concentration by approximately 40% in rodents [5]. Genetic variants and/or combinations of small-nucleotide polymorphisms in the human components of the IIS pathway correlate with low IGF-1 plasma levels in centenarians [7].

Nonetheless, a recent study with modified rodents lacking mTORC2 adipocyte activity questions the idea that improved insulin sensitivity is the mechanism through which CR increases health span and lifespan. Rodents without mTORC2 adipocyte activity, which is needed for the improvement of insulin sensibility in CR, had displayed an increase in their health and lifespan when exposed to CR [9].

Studies with non-human primates are also contributing to support the benefits of CR. One investigation shows how 30% CR drastically reduces the incidence of glucose intolerance, cardiovascular disease, and cancer in primates [10,11]. Another study showed that CR slowed down age-related sarcopenia, hearing loss, and brain atrophy in several subcortical regions [10,11]. In *Rhesus* monkeys, young-onset CR reduces the risk of developing and dying of cardiovascular diseases by at least 50% [3].

Some insights about the effects of CR without malnutrition in humans can be taken out of population studies. During World War 1, the Danish population was forced to reduce food consumption for 2 years, but with adequate consumption of whole-grain cereals, vegetables, and milk, which led to a 34% reduction in death rates. During World War 2, Oslo citizens also underwent a forced 20% CR without malnutrition for 4 years, and in this time, mortality dropped by 30% compared to the pre-war level [6].

There is evidence that CR exerts a powerful anti-inflammatory effect in humans, protecting against atherosclerotic risk factors, and resulting in less carotid intima-media thickening, improved left ventricular diastolic function, and increased beneficial heart rate variability [12,13]. Various metabolic and neuroendocrine mechanisms are responsible for this CR-mediated anti-inflammatory effect, including low adiposity and reduced secretion of proinflammatory adipokines and cytokines, reduced plasma glucose and advanced glycation end-product concentrations, increased cortisol and ghrelin production, and increased parasympathetic tone [14].

Despite there is no evidence at the moment to support the idea that long-term CR with adequate nutrition extends maximal lifespan in humans, it is known that long-term CR leads to metabolic and hormonal changes seen in rodents, including reduced body temperature and resting metabolic rate, reduced markers of oxidative stress and reduced fasting insulin levels [5].

### 2.2. Calorie Restriction and the Effects on Insulin Sensitivity

A recent multicenter randomized control trial, CALERIE 2 [3], evaluated the cardiometabolic risk factor responses to CR diet for 2 years in 220 young and middle-aged (21 to 50 years), healthy non-obese men and women. Participants were randomized to a 25% CR group or an ad libitum (AL) control group. Over the 2 years, the CR group achieved 11.9% CR and a sustained 10% weight loss, of which 71% was a fat mass loss. CR caused a significant and persistent reduction of all measured cardiometabolic risk factors, including Low-density Lipoprotein cholesterol (LDLc), total cholesterol to High-density Lipoprotein cholesterol (HDLc) ratio, systolic and diastolic blood pressure (BP). Additionally, CR resulted in a significant improvement in C-reactive protein (CRP), glucose tolerance, insulin sensitivity index, and MS score relative to control. A secondary analysis revealed the responses to be robust after controlling for relative weight loss changes [3]. Insulin was measured by chemiluminescent immunoassay and IR was calculated using the homeostatic model assessment of IR (HOMA-IR). Insulin response was calculated at the ratio of change in plasma insulin from baseline to 30 min to the change in plasma glucose over the same time. Insulin sensitivity was calculated as 1/fasting insulin. Fasting and area under the curve (AUC) insulin were both significantly reduced in the CR group as compared with the AL group at 1 and 2 years. Fasting glucose was significantly reduced by CR at year 1, but not at year 2. CR improved insulin sensitivity: HOMA-IR reduced; insulin response increased (at 2 years only); insulin sensitivity index increased. No significant reduction in the AUC-glucose was observed. Usually, glucose plasma concentrations are within the normal, non-diagnostic range, until two to five years before the diagnosis of T2DM when a rapid deterioration of insulin secretion and a parallel elevation of glycemia occur [3]. Contrarily, IR predicts the risk of T2DM as early as 30 years before diagnosis. In the CALERIE-2 trial, glucose tolerance did not significantly change at a point in the development of diabetes where it may still be in the normal range. However, insulin sensitivity greatly increased, while plasma fasting glucose and glucose-stimulated insulin concentration reduced. These findings suggest the potential for the significant cardiovascular advantage of practicing moderate CR in young and middle-aged healthy individuals, and possible long-term population health benefits [3].

Regarding safety concerns with long-term CR in non-obese healthy individuals, a previous study [15] concluded that two-years of CR at levels achieved in CALERIE are safe and well-tolerated. No deaths were observed in the study. Participants in the CR group reported more non-serious events regarding the reproductive system and skin disorders, than participants in the control group, but none of the differences was statistically significant. In the CR group, the incidence of adverse events was statically higher in normal weight than in overweight participants, regarding nervous, musculoskeletal, and reproductive disorders [15]. However, the number of participants was small, so no definitive inferences about the effects of body mass index (BMI) could be made. Close monitoring for excessive bone loss and anemia is important. Bone mass measurement (BMD) loss by the CR group was unclear. CR group participants had an increased risk of significant decreases in hematocrit [15]. It must be remembered that CR must be practiced as a CR diet without malnutrition.

BIOSPHERE 2 study was an observational study in which eight members of a crew faced a forced 29% CR for 18 months, and they experienced marked reductions in levels of insulin and glucose concentrations (91.9 to 73.9 mg/dL). CRON Study was also an observational study in which multiple members of the Calorie Restriction Society followed a regimen of self-imposed CR (1800 kcal/d) with optimal nutrition for an average of 15 years. Fasting glucose and insulin were remarkably low, and insulin sensitivity was improved [6].

Several mechanisms have been proposed to explain the effects of CR on glucose metabolism. Reduced energy intake reduces pancreatic cell apoptosis. Improved insulin sensitivity increases the expression of SIRT-1, which is probably linked to hepatic glucogenic/glycolytic pathways, increasing hepatic glucose output. SIRT-1 also enhances endothelial nitric oxide synthase (eNOS) activity, whose increase is in turn related to the reduction of oxidative stress in endothelial cells caused by CR [8]. CR increases the levels of adiponectin in humans, which is inversely related to body weight, adiposity, and IR. Adiponectin modulates insulin activity and also reduces insulin levels and beta-cell dysfunction. CR also positively modulates the secretion of adipocyte cytokines by decreasing the secretion of proinflammatory mediators and the development of a pro-inflammatory phenotype in white adipose tissue [8].

A three-week non-controlled CR intervention with a 40% energy deficit in 41 non-obese adults stratified the subjects into two enterotypes, according to their baseline microbial composition, in Bacteroides subjects and *Prevotella* subjects [16]. *Prevotella* subjects exhibited a significantly higher BMI loss ratio than Bacteroides subjects, showing that subjects with different baseline enterotypes can respond differently to CR diet and that pre-intervention gut microbial composition could well predict CR-induced BMI loss. No changes in gut microbial composition were observed by the end of the study [16].

A 2012 review article assembles the evidence regarding the beneficial effect of CR on age-related cardiovascular disease [17]. The explained mechanisms for reduced atherosclerosis and improved insulin sensitivity are: decreases the accumulation of oxidized lipids and reduces oxidative stress in the arterial wall; decreases inflammation (i.e., TNF-α, interleukin 6 (IL-6), CRP); decreases blood glucose and lipids (i.e., triglycerides (TG), cholesterol) [17]. CR can significantly improve heart rate variability and arterial stiffness and dilatation, normalizing BP values. BP improvement may be related to insulin sensitivity and the increase in nitric oxide (NO) production. By increasing insulin sensibility, CR improves BP through multiple mechanisms. Hyperinsulinemia leads to peripheral vasoconstriction, increases the reabsorption of sodium by the kidney, stimulates the renin-angiotensin II axis, increases smooth muscle hypertrophy, and stimulates the sympathetic nervous system, leading to increased BP. Hyperinsulinemia also increases the production of endothelin 1, which enhances BP. Hyperinsulinemia and IR lead to a dysfunctional inositol 3-kinase-dependent signaling pathway, decreasing NO production [18].

### 2.3. Calorie Restriction, Interactions between Insulin and Exercise

Prediabetes starts as a skeletal muscle, liver, and/or adipose tissue IR, that in time promotes oversecretion of insulin and results in pancreatic exhaustion with hyperglycemia. Skeletal muscle IR is primarily responsible for impaired glucose tolerance (IGT; 2 h postprandial glucose > 140 mg/dL) while hepatic IR is mainly responsible for impaired fasting glucose (IFG; fasting plasma glucose > 100 mg/dL) [19]. Prediabetic individuals with IFG+IGT have lower insulin sensitivity responses to high-volume/high-intensity exercise when compared to IFG or IGT alone. Thus, a recent study hypothesized that CR could improve the insulin sensitivity response to exercise in prediabetic individuals [19]. The energy deficit that follows immediately after exercise has an important role in the benefits of exercise on insulin sensitivity. Postprandial insulin levels decrease more when the energy deficit is maintained after exercise, compared to when calories are restored by ingestion. This is observed in short-duration studies as well as in long-duration studies (several months to 1 year). Increased fasting fat oxidation after exercise with calorie and/or carbohydrate restriction was observed, suggesting that mitochondrial reliance on fat may contribute to improved insulin sensitivity. Other studies with healthy individuals and individuals with obesity and/or T2DM demonstrated that when after exercise the calorie ingestion compensates the calories that were spent, insulin sensitivity does not statistically improve, reinforcing the idea that the effect of exercise on insulin sensitivity is due to the calorie deficit [19,20,21]. Hence, CR associated with exercise can have a greater impact on insulin sensitivity than exercise alone.

A 5-month randomized controlled trial in 126 older (>65 years) overweight and obese men and women also studied the efficacy of adding CR for weight loss to resistance training (RT) on MS. Compared to RT, RT+CR resulted in a decrease in body mass, VLDL cholesterol, TG, and systolic and diastolic pressures. Yet, there was no significant difference between groups on insulin sensitivity, determined by HOMA-IR. These results are aligned with previous studies with similar individuals but go against studies done with younger individuals [22].

## 3. Ketosis-Inducing Diets, Definition, Types and Effects

Ketosis-inducing diets represent a group of diets that induce the production of ketones, comprising: diets with low carbohydrate content, normoproteic content and high-fat content, such as the classic ketogenic diet (CKD) and the fasting mimicking diet (FMD); and diets in which the individual restricts the eating period, having intermittent ketosis, such as intermittent fasting (IF).

Carbohydrates are the main source of energy used by the organism. When less than 50 g of carbohydrates are ingested per day, the insulin secretion decreases significantly, and the body enters a catabolic state. Glycogen stores are depleted, and the organism is forced to enter metabolic changes, mainly: gluconeogenesis (endogenous glucose production) and ketogenesis (ketone bodies production by the liver) [23]. Ketone bodies (KB) are lipid driven molecules (beta-oxidation), and they act as alternative substrates for energy production during fasting and low-carb diets [24]. KB act as powerful anorexigenic agents, reducing cerebral neuropeptide Y, maintaining cholecystokinin (CCK) meal response and decreasing circulating ghrelin, with a general reduction of perceived hunger and food intake, increasing weight loss and diet tolerability [25].

Physiological ketosis is considered safe since KB are produced in small concentrations and without alterations in plasma pH. As seen in Table 1, physiological ketosis differs from ketoacidosis, a life-threatening condition in which KB are produced in extremely higher concentrations, modifying plasma pH leading to acidosis [23]. Blood levels of KB in healthy people do not exceed 8 mmol/L precisely because the central nervous system (CNS) efficiently uses these molecules for energy in place of glucose [26].

International consensus establishes carbohydrates as the base of the food pyramid for healthy nutrition and advocates that the best approach to lose weight is through reduction of calories, mainly from fat sources. Alongside, there is a belief that ketogenic diets may lead to the development of various diseases. Nevertheless, several studies have shown that ketosis-inducing diets are healthier, as they help to preserve muscle mass while increasing fat loss, reduce appetite, induce metabolic thermogenesis, promote a non-atherogenic lipid profile, reduce BP, and reduce IR improving glucose and insulin levels. Contrarily, diets based on carbohydrates may be linked to low levels of HDLc, higher levels of TG, LDLc and total cholesterol, and an increased risk for developing MS, T2DM, hypertension and cancer [27].

### 3.1. Ketogenic Diet, Definition and Effects

Wilder introduced the term “ketogenic diet” and tried it for the first time to treat epilepsy in 1921, as low-carb and high-fat, resulting in a normocaloric diet. For a while, the ketogenic diet (KD) enjoyed a place in medicine as a therapeutic diet for pediatric epilepsy and was widely used until its popularity ceased with the introduction of antiepileptic agents [23].

Recently the KD has regained reputation, even in the scientific world, as a non-pharmacological therapeutic strategy for weight management and metabolic regulation. This dietary approach is an evolution of the initial KD, applying CR.

The CKD consists of a diet high in fat, moderate in protein and low in carbohydrates, with CR. This diet mimics the fasting state, altering metabolism to use fats as a primary source of fuel; fatty acid catabolism in the liver produces KB, which induce ketosis [28].

The resurgence of the KD as a form of rapid weight loss is a relatively new concept that has been shown to be quite effective, at least in the short term [23]. In recent years, alternative KD protocols and other characteristics have emerged, in addition to the composition of macronutrients, which have been increasingly recognized as important factors for long-term adherence and effectiveness of KD. These characteristics include fatty acid composition and nutrient density [29]. There are currently four main ketogenic dietary therapies (KDTs) [30,31]: the classic KD (as mentioned above), the modified Atkins diet (MAD) (for obesity treatment) [32], the medium-chain triglyceride (MCT) (control of metabolic diseases) diet [33], and the low glycemic index treatment (LGIT) (treatment of intractable epilepsy) [30,34,35,36].

Unfortunately, in the literature, no clear definition of KD is provided, and many studies define a KD as any diet that leads to an increase in ketones in the blood, for example, diets in which no more than 50% of the total calories come from fat [29].

A 2020 meta-analysis evaluated the effects of ketogenic diet (KD) in T2DM, and fasting glucose decreased 23.24 mg/dL and glycated hemoglobin decreased 1.07 [37]. A consensus paper of 2019 by the American Diabetes Association concluded that diets with low carbohydrate content (including those leading to physiological ketosis) “are among the most studied dietary patterns for T2DM” and that these “dietary patterns, especially very low carbohydrates have been shown to reduce glycated hemoglobin (HbA1c) and the need for anti-hyperglycemic drugs” [38]. A case report highlights a 65-year-old woman who had a 26-year history of doubly diagnosed T2DM and severe depressive disorder, for whom a KD was prescribed. The results indicated better insulin sensitivity assessed by HOMA-IR, glycemic control measured via HbA1c, reduced cardiovascular risk through the TG/HDLc ratio and improved depressive symptoms with increased self-efficacy monitored by Patient Health Questionnaire 9 (PHQ-9) and General Self-Efficacy Questionnaire (GSE) / Metabolic Syndrome Compliance Questionnaire (MSC) [39].

In a prospective study with 31 obese subjects, a KD called “Spanish ketogenic Mediterranean diet” (SKMD) was evaluated for 12 weeks [27]. SKMD included virgin olive oil as the principal source of fat, moderate red wine intake, green vegetables, and salads as the main source of carbohydrates and fish as the main source of proteins. There was an extremely significant (*p* < 0.0001) reduction in body weight, body mass index, systolic BP, diastolic BP, total cholesterol, triacylglycerols and glucose (109.81 mg/dL → 93.33 mg/dL). This improvement in fasting glycemia could be explained by the following points: (1) A low carbohydrate diet reduces fasting glucose levels, even independently of the weight loss. (2) Monounsaturated fatty acids rich diet prevents IR induced by a carbohydrate-rich diet in insulin-resistant subjects. (3) Docosahexaenoic acid—eicosatetraenoic acid (DHA-EPA) also improves insulin sensitivity [27].

Exogenous ketone supplements (EKS) such as ketone esters and ketone salts—β-hydroxybutyrate—have recently been available. When taken in fasting periods these supplements induce the ketogenic environment of fasting more quickly. A study found that EKS reduced appetite for more than four hours [40]. The adverse effects usually range from gastrointestinal discomfort to diarrhea [41]. It is a strategy that is still very recent, and evidence of efficacy and safety are still lacking [24,42,43].

KD also showed modulating effects over the microbiota [44]. Stool samples form obese and overweight non-diabetic men after 4 weeks of KD showed phylum-level reductions in the relative abundance of the Firmicutes and a major reduction in the Actinobacteria, mainly in *Bifidobacterium*, while Bacteroidetes were modestly increased [45]. Previous studies with carbohydrate-inclusive high fat diets (HFD), that have no increase in ketosis markers, showed opposing trends on microbiome composition [46]. In KD, the relative abundance of Actinobacteria was dose dependent, decreasing with a decrease in carbohydrates, suggesting a causative link between KB and Actinobacteria. In vitro cultures of B. adolescentis treated with β-hydroxybutyrate had a reduced growth, but neutralization of the media pH reversed the effects [45]. B. adolescentis was previously associated with expansion of pro-inflammatory Th17 cells, especially in the context of multiple autoimmune disorders [47].

### 3.2. Intermittent Fasting, Definition and Effects

Fasting is an ancient practice followed in a variety of different formats by people around the world and in different religious disciplines, including Islam, Christianity, Judaism and Hinduism [31].

Fasting can be defined as voluntary abstinence or reduction of some or all foods, drinks or both (absolute) for a period that lasts between 12 h to 3 weeks, whether in a short-term, long-term/prolonged or a flashing pattern [31]. Currently, it is gaining more popularity [48] and is being investigated as a potential non-pharmacological intervention to improve health and increase longevity [49].

Intermittent fasting (IF) includes various practices, and study’s protocols vary widely [50], with alternating periods of feeding and periods of fasting that can go up to 24 h from one to four days a week [50]. Table 2 and Figure 1 define the various terms used to describe the different types of IF regimes discussed in this review, and this terminology will be used likewise, even though different studies may use different terms.

The fasting period also works as a resting and repairing period for the organism to be prepared to receive food when it becomes available [53]. The circadian clock interacts intimately with nutrient sensing pathways. Frequent eating and the absence of a defined fasting period disturb the normal counter-regulatory metabolic state that occurs during fasting.

Proper fasting may allow optimization of the organs’ peripheral clocks such as those in the liver, adipose and skeletal tissues. Dysregulation of these systems increases the risk of chronic diseases, as evidenced by the higher rate of cardiometabolic diseases in shift workers. One circadian example relevant to IF is the decreasing insulin levels later in the day. Late dinners are associated with higher postprandial glucose levels than daytime meals, increasing the risk of diabetes. In humans, circadian misalignment increases insulin resistance after only 3 days. Nighttime eating decreases both quality and quantity of sleep, which also leads to increased insulin resistance, obesity, and cardiovascular disease [60]. Recent studies have shown that the intestinal microbiota and its metabolites present a diurnal rhythm that responds predominantly to the feeding/fasting cycle (Figure 2). Sleep quality and duration may be an important target to support the healthy composition of the intestinal microbiota [65].

During feeding, activation of the insulin phospho-protein kinase (pAKT)-mechanistic target of rapamycin (mTOR) pathway drives downstream gene activities that promote anabolic processes. In contrast, a few hours of fasting activate AMPK, which triggers repair and catabolic processes [53]. Regular periods of fasting can provide physiological benefits, such as reduced inflammation, improved circadian rhythmicity, increased autophagy and resistance to stress, and modulation of the intestinal microbiota [57].

Studies in normal and overweight subjects have shown efficacy of IF primarily for weight loss and improvements in several health indicators, including IR and reductions in risk factors for cardiovascular disease [51]. A study with three patients referred to the Intensive Dietary Management clinic in Toronto, Canada, for insulin dependent T2DM, showed a reversion of IR and improvement on glycemic control, leading to an interruption of insulin therapy. In addition, subjects lost significant body weight, reduced waist circumference and HbA1c [66]. A 5-week randomized controlled trial in pre-diabetic men showed an improvement in insulin levels and insulin sensitivity, increased β cell response, and lower BP and oxidative stress levels [57].

The most immediate risk of intermittent fasting is the possible hypoglycemia in patients who are on antidiabetic medications that can cause hypoglycemia [58,67]. With long-term IF, one needs to be concerned about protein malnutrition if patients are not cognizant to maintain adequate protein intake on eating days. Vitamin and mineral malnutrition can also occur [58].

Other risks include a variety of potential harms related to insufficient energy intake and some due to dehydration. These include safety events that may occur among anyone who engages in intermittent fasting, regardless of whether they have diabetes. Such adverse events may include dizziness, nausea, insomnia, syncope, falls, migraine headache, weakness that limits daily activities, and excessive hunger pangs [58]. That is why it is encouraged that the introduction of this type of nutritional strategy is done with the accompaniment of a professional, such as a doctor and/or a nutritionist.

### 3.3. Very-Low-Calorie Ketogenic Diet, Definition and Effects

Very-low-calorie ketogenic diet (VLCKD) is a nutritional intervention that mimics fasting characterized by low-fat (10 g/day of olive oil) and low-carbohydrates (usually lower than 30 g/day from a vegetable source) formulations and a high-biological-value protein content of 0.8 to 1.2 g/kg of ideal body to prevent lean mass loss, starting with a total daily energy intake < 800 kcal for about 2 weeks [25,68,69]. The weight-loss program of VLCKD has an initial ketogenic period (8–12 weeks) and a gradual reintroduction of daily calories [25].

VLCKD has been associated with a dramatic reduction in insulin and oral glucose-lowering medication requirements, including total remission of diabetes [44]. VLCKD studies in young obese subjects have also shown improvements in BP, lipid profile, fasting insulin and glucose, and insulin sensitivity [25].

A 2019 systematic review and meta-analysis evaluated the efficacy and safety of VLCKD in overweight and obese patients [70]. VLCKD was associated with weight loss (−10.0 kg in studies up to 4 weeks and −15.6 kg in studies of at least 4 weeks), which was sustained in the 2 years follow-up. VLCKD was also associated with reductions of BMI, waist circumference, HbA1c, total cholesterol, TG, AST, ALT, GGT, systolic and diastolic BP. No changes were found in LDLc, HDLc, serum creatinine and serum potassium [70].

A study evaluated the effects of 8 weeks of VLCKD on body composition and resting energy expenditure in the short-term reversal of T2DM [71]. VLCKD led to saving of lean mass, reduction of abdominal fat mass, restoration of metabolic flexibility, maintenance of resting energy expenditure. Additionally, VLCKD led to a reduction in fasting glycemia (−39.7%) and a glycated hemoglobin below 6.5%, with subjects returning to within normal ranges, as previous studies had shown. These results indicate a short-term remission of T2DM, but the long-term effects with the reintroduction of carbohydrates were not assessed [71].

The Italian Society of Endocrinology formulated evidence-based recommendations for the use of VLCKD in different clinical settings [25]. Strong VLCKD recommendations, all with moderate evidence, are for: severe obesity; management of severe obesity before bariatric surgery; sarcopenic obesity; obesity associated with T2DM (preserved β-cell function); obesity associated with hypertriglyceridemia; obesity associated with hypertension; pediatric obesity associated with epilepsy and/or with a high level of IR and/or comorbidities, not responsive to standardized diet [25].

### 3.4. Fasting Mimicking Diet, Definition and Effects

The FMD is a plant-based diet designed to attain fasting-like effects on the serum levels of IGF-1, Insulin-like Growth Factor-Binding Protein 1 (IGFBP-1), glucose, and KB while providing both macro-and micronutrients to minimize the burden of fasting and adverse effects [72].

The Fasting Mimicking Program (FMP) is a program based on healthy natural products and ingredients, generating a hormonal and metabolic response that mimics fasting [48,49]. The FMD diet consists of a 5-day regimen: day 1 diet of the diet supplies ~1090 kcal (10% protein, 56% fat, 34% carbohydrate), days 2 to 5 are identical in formulation and provide 725 kcal (9% protein, 44% fat, 47% carbohydrate) [49]. Is composed of plant-based ingredients, such as energy bars, vegetable soups, glycerol-based energy drinks, seaweed oil, herbal tea and other snacks: olives, cabbage biscuits and chocolate bars [49].

A randomized study involved the participation of 100 healthy participants from the United States, who were divided into two groups (control and FMD—low in calories, sugars and proteins, but rich in unsaturated fats). Individuals who followed 3 months of the unrestricted diet were compared to subjects who consumed FMD for 5 consecutive days per month for 3 months. The results show that three cycles of FMD reduced body weight, trunk and total body fat; low BP; and decreased IGF-1. No serious adverse effects have been reported [72]. Another 2015 study, using a similar methodology, also observed considerable weight loss, in addition to decreased visceral fat, reduced inflammation, reduced incidence of cancer and skin lesions, rejuvenated immune system and reduced incidence of cardiovascular disease and diabetes [49]. 

Another 2019 review article concludes that protein restriction and a diet that mimics fasting (FMD) together have the potential to play an important complementary role in medicine, promoting the prevention and treatment of diseases (such as diabetes, cardiovascular diseases) and cancer) and delaying the aging process, at least in part, by stimulating stem cell-based regeneration during periods of normal food intake after periodic FMD cycles [73].

Recent studies are showing benefits of fasting as an anticancer strategy [48,74,75,76,77,78,79]. The justification for this concept is that fasting causes a differential response to stress in the context of unfavorable conditions, strengthening the survival of normal cells, while killing cancer cells. Among the diverse dietary deprivation interventions, FMD has been standing out [75].

In a 2020 randomized study with 131 HER2-negative stage II/III breast cancer patients without diabetes and a BMI over 18 kg/m2, participants received FMD or their regular diet for 3 days before and during neoadjuvant chemotherapy [76]. Protocol analysis reveals that the Miller & Payne 4/5 pathological response, indicating 90–100% loss of tumor cells, is more likely to occur in patients using FMD. In addition, FMD significantly reduces DNA damage induced by chemotherapy in T lymphocytes. These positive findings encourage further exploration of the benefits of fasting and FMD in cancer therapy [76].

### 3.5. Ketogenic-Inducing Diets and Their Effects on Neurodegenerative Diseases

There is evidence that KB have a neuroprotective impact on aging brain cells, enhance mitochondrial function, an reduce the expressions of inflammatory and apoptotic mediators [80], suggesting a positive impact of KD on the prevention and treatment of neurodegenerative disorders. Emerging data suggests that KD may be useful in amyotrophic lateral sclerosis (ALS), Alzheimer’s disease (AD), Parkinson’s disease (PD), and some mitochondriopathies [80,81,82,83]. These diseases share common pathogenic mechanisms that could explain the effects of KD.

Once KB reach a concentration of about 4 mmol/L they start to be utilized as energy source by the CNS [84]. Compared to glucose, KB can produce a higher quantity of energy due to the changes in mitochondrial ATP production that they induce [84]. Therefore, KD provides an efficient source of energy for the treatment of certain neurodegenerative diseases characterized by focal brain hypometabolism, such as AD and PD [82,85]. KB can provide a source of cytoplasmic acetyl-CoA that can attenuate the lowering of acetyl choline typical of AD. Moreover, AD is associated with metabolic dysregulation and IR [86,87,88], that may improve with KD.

KD decreases the oxidative damage associated with metabolic stress. Compared to glucose metabolism, KB generate lower levels of oxidative stress in the brain and a greater cellular energy output and antioxidant capacity [81,82]. Moreover, ketosis can increase glutathione peroxidase in hippocampal cells and in general decreases mitochondrial ROS production [89]. KB reduce the production of free radicals by improving the efficiency of the mitochondrial respiratory chain complex. The mitochondrial dysfunction results in diminished energy generation and it can increase beta-amyloid accumulation and tau protein dysfunction [80]. The restriction of calories increases the levels of neuroprotective factors, such as neurotrophins and molecular chaperones [81].

KD increases the mitochondrial biogenesis pathways, leading to increased generation of ATP and enhanced energy reserves, stabilizing synaptic activity and improving neuronal metabolism [80,82]. Also, KB can bypass the defect in complex I activity implicated in some neurological diseases, like PD and ALS [90]. In cultured neurons treated with pharmacologic agents blocking complex I, an addition of KB restores the function of the complex [91]. KD protected dopaminergic neurons of the substantia nigra against 6- hydroxydopamine neurotoxicity in a rat model of PD [92]. In an ALS mouse model KD led to a higher motor neuron survival and an improvement in motor function [93]. Also, supplementation with a precursor of KB improved mitochondrial function and motor neuron count in an ASL mouse model [94].

On AD, an in vitro study demonstrated that the addition of KB protects the hippocampal neurons form β-amyloid toxicity [95]. Also, KD was found to reduce the volumes of solved beta-amyloid in murine brains [96], and long-term administration of KB esters to mice improved their cognitive functions and reduced β-amyloid and highly phosphorylated tau proteins in the brain [95]. β-Amyloid induces a decline in mitochondrial function [81].

Studies with human participants demonstrated a reduction of disease symptoms after application of KD in AD [97,98,99,100,101,102,103]. Studies have demonstrated that a supplementation with medium chain triglycerides that induce an increase of KB improved performance in the AD assessment cognitive scale with a direct correlation between ketone concentration and cognitive improvement [97,98]. Similar results of improved cognitive function with supplementation of medium chain triglycerides were observed in other studies [99,100]. The best results of KD treatment are expected in early presymptomatic stages of AD [80].

Regarding PD, in a study with 28 PD patients that prepared a “hyperketogenic” diet at home and adhere to it for 28 days, the high level of KB was related to an improvement in the Unified Parkinson’s Disease Rating Scale scores [104].

However, the application of the KD to elderly people raises certain concerns. Persons with neurodegenerative diseases are at risk of malnutrition. As KD leads to a reduced appetite and may be have gastrointestinal side effects, it can worsen this situation. Further research is needed to evaluate the suitability of the ketogenic diet in this population [81].

## 4. Calorie Restriction and Ketosis-Inducing Diets in Cancer

Most cancer cells, contrarily to normal differentiated cells, rely on aerobic glycolysis to generate the energy needed, a phenomenon termed the Warburg effect, the seventh feature of cancer cells [105]. Therefore, cancer cells exhibit high rates of glucose uptake and lactic acid production, even in the presence of oxygen, preferring aerobic glycolysis to oxidative phosphorylation. This metabolic shift presents bioenergetic and biosynthetic advantages to proliferating cells by increasing non-oxidative adenosine triphosphate (ATP) production and generating metabolic intermediates from glucose [106]. Glutamine represents an additional metabolite catabolized by tumor cells and utilized for biosynthetic processes [107]. Additionally, aerobic glycolysis and glutaminolysis contribute to the chemoresistance of cancer cells [108].

Several compounds are known to mediate this effect: increased expression of glucose transporters and thus increased glucose uptake; increased pentose phosphate pathway-catalyzed nicotinamide adenine dinucleotide phosphate (NADPH) production; altered activity of glycolytic or glycolysis-related enzymes; and increased lactate production [105].

Diet-based strategies can be used additionally to chemotherapy and radiotherapy to target these metabolic processes. CR and KD reduce glucose levels, eliminating the benefits of glycolysis to cancer cells [105], and shutting off the energy source of the cancer, as the decrease in glycolytic ATP production cannot be compensated for via oxidative phosphorylation [109]. These strategies have been applied in both preclinical and clinical studies. However, robust evidence is still lacking, and more studies are needed [105].

By reducing the plasmatic levels of glucose, CR and KD act on common target molecular pathways, including Phosphoinositide 3-kinase (PI3K), mTORC and AMPK [105]. CR activates the Nuclear factor erythroid 2-related factor 2 (Nrf2) gene, which counteracts the insulin/IGF-1/PI3K/mTOR pathway and promote mitochondrial function [105]. The lower levels of circulating glucose lower insulin levels and increase IGFBP-1, which decreases the bioavailability of IGF-1 [110]. Contrarily, the activation of the insulin receptor stimulates Ras and mitogen-activated protein Kinase (MAPK) cascade, promoting cell proliferation, works through the PI3K pathway to promote cell survival, through AKT and mTOR, and induces vascular endothelial growth factor (VEGF) [111], as seen in Figure 3.

The impact of CR on cancer suppression has been replicated in studies of brain, prostate, and breast tumors [105,112,113]. In animal models CR increased radiation efficacy in breast cancer [114]. In animal models KD has been shown to delay human gastric cancer cell growth [115] and prostate cancer [116].

KD appears to have pleiotropic metabolic, genetic, and immune effects on cancer [111], and KB show anti-cancer properties in vitro and in vivo [117,118,119,120], although the benefits and possible risks may be cancer type-specific, with negative impacts in certain cancers [111,121,122]. The role of ketosis in cancer is related to the following mechanisms: (1) reduction of glucose and insulin, decreasing glucose availability to the tumor; (2) modulation of oxidative stress; (3) reduction of inflammation; (4) enhancement of anti-tumor immunity; (5) alteration of gene expressions, reducing the expressions of genes involved in the hypoxic response, angiogenesis, and invasive potential; and (6) sensitization of tumors to standard of care and adjuvant therapies [111]. While protecting healthy tissues against oxidative stress, KD appears to lower basal oxidative stress within tumors but enhance oxidative stress-induced damage in response to chemotherapy and radiation, which work in part by enhancing reactive oxygen species (ROS) production [111]. The results may be due to reduced metabolic substrates necessary for a significant antioxidant response [111].

In a clinical trial, 10 patients with end-stage cancer receiving KD for 28 days exhibited a significant decrease in blood insulin, and a positive correlation between ketosis degree and therapeutic response [123]. As exogenous ketones supplements may also suppress glucose and insulin systematically, they may have a role in cancer therapy, and may support dietary compliance [111].

Similarly, fasting, that induces physiologic ketosis, increases therapeutic response to chemotherapy in melanoma, glioma, colon carcinoma and breast cancer cells, as well as in a mouse model of neuroblastoma and colon carcinoma [124,125]. In cancer patients fasting is not associated with major side effects and may reduce several of the side effects associated to chemotherapy [126], as well as protect normal cells against chemotoxicity and increase the effectiveness of chemotherapeutic [127]. The metabolic alterations observed in tumor cells are often related to induction of PI3K- AKT signaling [124]. Fasting leads to reduced availability of glucose and aminoacids in the extracellular space, reducing the expression of the glucose transporters and leading to a general reduction in the glycolytic rate [124]. Inhibition of glucose utilization is accompanied by a down-regulatory effect on the glutamine transporters and glutaminase, which catalyze glutamine catabolism and provide cancer cells with biosynthetic precursors for aminoacids and DNA synthesis [124,128].

## 5. Calorie Restriction vs. Ketosis-Inducing Diets

Although calorie restriction is a robust tool to preserve and enhance health, different dietary approaches may be more beneficial regarding specific diseases and other individual features. As we have been discussing, there are a variety of protocols and diet approaches to induce ketosis, with different positive and negative impacts that must be individually considered. Hence, comparing CR to ketosis-inducing diets requires a separate assessment of different protocols. The most studied comparison is between CR and intermittent fasting, while trials assessing other diets are lacking in the literature.

### 5.1. Continuous Calorie Restriction vs. Intermittent Calorie Restriction

Some of the most important arguments against long-term CR are its difficult implementation to some individuals and the risk of malnutrition. Intermittent fasting seems to reduce the risk of malnutrition and to be easier to follow by some individuals [8]. Yet, it is important to consider behavioral modifications, such as binge eating, following a fasting period [8].

Another important factor is that CR usually reduces fat and lean body mass, while intermittent fasting diminishes fat mass while preserving lean body mass [8].

A randomized study from 2017 compared intermittent energy restriction (2 consecutive days a week with very low energy consumption (25%)) to continuous CR on overweight and obese participants and measured the impact of these diets on postprandial glucose and lipid metabolism following matched weight loss [129]. The matching of the weight loss was important as it leads to improved glucose and lipid homeostasis by itself. The study found no statistical difference in the time to achieve a 5% weight loss between groups. For postprandial measures, neither diet significantly altered postprandial glycemia, however, insulinemia was reduced comparatively in both groups. The reduction in C-peptide and TG tended to be greater following the intermittent CR diet. C-peptide undergoes negligible extraction by the liver and constant peripheral clearance, making it a more direct marker of insulin secretion than circulating insulin. This suggests that intermittent energy restriction diets may reduce insulin secretion over the first 2 h hours of the postprandial period, which has resulted in postprandial C-peptide and insulinemia reduction. Continuous energy restriction only caused postprandial insulinemia reduction, suggesting an increase in hepatic insulin clearance [129].

Previous studies that compared intermittent calorie restriction (ICR) to continuous calorie restriction (CCR) in overweight women showed a greater benefit of ICR in comparison to CCR on hepatic insulin sensitivity (fasting insulin and IR) [130,131]. However, these studies had not controlled for the extent of weight loss.

A study from 2019 compared the effects of ADMF to daily CR on body weight and glucoregulatory factors in adults with overweight/obesity and IR, for 12 months [63]. ADMF consisted of 25% energy needs on fast days and 125% on alternating feast days, while CR consisted of 75% energy needs every day. There was also a control group that maintained habitual energy consumption and activity. Bodyweight, fat mass, and BMI decreased similarly by ADMF and CR, but ADMF produced significantly decreases (*p* < 0.05) in fasting insulin and IR when compared to CR and controls by month 6 and month 12. Fasting glucose levels were not significantly different between ADMF, CR, and control groups by months 6 and 12 [63].

A previous randomized trial also compared the effects of ADMF vs. daily CR on 100 healthy obese adults [64]. ADMF also consisted of 25% energy needs on fast days and 125% on alternating feast days, CR consisted of 75% energy needs every day, and there was also a no-intervention control. The trial involved a 6-month weight-loss phase followed by a 6-month weight-maintenance phase. In this study, ADMF did not produce superior adherence, weight loss, weight maintenance, or cardioprotection vs. daily CR. Mean weight loss was similar in the ADMF group and CR group at months 6 and 12. There were no significant differences between the intervention groups in BP, heart rate, TG, fasting glucose, fasting insulin, IR, CRP, or homocysteine concentrations at months 6 or 12. Mean HDLc levels at month 6 significantly increased in the ADMF group, but not at month 12, relative to those in the daily CR group. Mean LDLc levels were significantly elevated by month 12 in the ADMF group compared with those in the daily CR group. The dropout rate was highest in the ADMF group (38%) vs. CR group (29%) and control group (26%), and more participants in the ADMF group withdrew due to dissatisfaction with diet, suggesting that ADMF may be less sustainable in the long-term [64]. No recent studies on lean and healthy individuals were founded during the research, reinforcing the need for further investigation in this area.

### 5.2. Calorie Restriction vs. Very-Low-Carbohydrate Ketogenic Diet

There is evidence that carbohydrate-restricted diets may promote greater weight loss than conventional energy-restricted low-fat diets (LFD) [132,133], yet a clinical trial that assigned individuals to diets ranging from 35% to 65% of dietary carbohydrate content showed no difference in weight loss between interventions [134]. This may be explained by the studied level of carbohydrate intake, as evidence suggests that greater dietary carbohydrate restrictions lead to greater weight loss [132].

Very-low-carbohydrate ketogenic diet is a diet with no more than 50 g carbohydrates/day. A recent meta-analysis aimed to investigate whether individuals assigned to a very-low-carbohydrate ketogenic diet achieve better long-term body weight and cardiovascular risk factor management when compared with individuals assigned to a conventional low-fat diet (a restricted-energy diet with less than 30% of energy from fat) [132]. Individuals assigned to the very-low-carbohydrate ketogenic diet revealed decreased bodyweight, TG and diastolic BP, while increased HDLc and LDLc. Attention must be paid to LDLc levels, as a previous investigation showed that high fat intake combined with carbohydrate restriction raises the levels of larger-sized LDLc, which are less atherogenic than the small dense LDLc [132]. Individuals assigned to a very-low-carbohydrate ketogenic diet also achieve a greater weight loss than those assigned to LFD in the long-term. Regarding fasting blood glucose, insulin, HbA1c and CRP, these analyses were performed in less than ten studies, and none of them showed statistically significant results. In the long-term and when compared with conventional therapy, the differences appear to be of little clinical significance, although statistically important [132]. A major concern of the very-low-carbohydrate ketogenic diet is the adherence of the individuals assigned to it.

## 6. Conclusions, Limitations and Future Research

Calorie restriction is the most robust intervention known until the moment to increase maximal lifespan and health span in large studies, with high evidence, biological plausibility and explained biochemical mechanisms [1]. The concepts of health span and lifespan implicitly include the prevention and the management of CNCD (cardiovascular disease, neurological degeneration of senescence and some cancers). The current review of the effects of CR and especially ketosis-inducing diets in those specific pathologies concludes that there is robust evidence on the benefits of these nutritional strategies as a complementary non-pharmacological treatment in the management of not only of cardiovascular disease, but also of cancer [105,111] and neurodegenerative disease, as Alzheimer’s disease [80,81,82].

Among the various mechanisms described are adaptive neuroendocrine responses, hormetic responses, modulation of the inflammatory response and the redox system. There are also described DNA repair mechanisms, defensive autophagic responses to the accumulation of damaged proteins and oxidized lipids, regulatory sense energy pathways, epigenetic control and gene expression [5]. Experimental studies demonstrated that insulin sensitization is the mechanism with the greatest impact on the development of CNCD [5,6].

Robust studies showed that CR improves crucial cardiometabolic risk parameters (LDLc; total cholesterol/HDLc; systolic and diastolic BP; CRP; glucose tolerance; HOMA-IR as an indicator of insulin sensitivity; metabolic score syndrome), especially after relative weight loss [3,12,13,17]. This response seems to be more significant after the 1st year of CR than at the end of the 2nd year of CR. We wonder whether it was the effect of the described reduction in the metabolic rate or whether the gradual loss of adherence also leads to a loss of the effectiveness of this long-term strategy. Nevertheless, the insulin response is higher at the end of the 2nd year and it seems to be a later response [3].

Studies from a month to a year of intervention show that the determining factor of the benefit of exercise in IR is the energy deficit that follows exercise, as when calories are not replaced, and the energy differential is maintained. The effects on increasing fat oxidation and improving insulin sensitivity are greater when conjugating CR with exercise [19,20,21,22].

Although CR represents a powerful tool to preserve health, different approaches in quantity and quality of calories eaten should be used when dealing with different patients’ treatment of specific diseases, such as metabolic diseases, cardiovascular diseases, and cancer. Also, patients age and personal preferences are important aspects to consider. 

Studying the concept of physiological ketosis, we are convinced that there is still a prejudice regarding ketosis as a possibly dangerous and potentiator of disease. Yet, recent studies have shown that physiological ketosis may have benefits that were not anticipated [27], such as preventing mass loss muscle, appetite control, induction of thermogenesis, and promotion of non-atherogenic lipid profile, in addition to superior improvement in the parameters of glycemic metabolism and cardiovascular risk, compared to the classic low-calorie diet based in carbohydrates [27]. There is proven evidence of the application of CKD for weight and MS management [38,39]. Also, SKMD [27] is well described with extreme significance (*p* < 0.0001) benefits for all the metabolic parameters studied in addition to weight loss.

The most studied comparison of CR diets is between CCR and ICR, while trials assessing other diets are lacking in the literature. Comparison studies indicate that ICR may have greater benefits compared to CCR [129,130,131]. Golbidi et al. [8] defend ICR due to its possible easier adherence in the medium/long-term compared to CCR, as it has a lower risk of malnutrition, better preservation of fat-free mass and less reduction in the metabolic rate [8]. Comparing CCR vs. ADMF diets [63,64] that accumulate the same energy supply at the end, there were no statistical differences between strategies in the time to achieve 5% of weight loss. Studies have shown an identical reduction in BMI and body composition but have described a more marked reduction in IR and fasting insulin with ADMF. Yet, another study showed that insulin decreased equally in CCR and ADMF, but the C-peptide reduction was more pronounced after ADMF, suggesting a reduction in postprandial insulin production [63,64].

Consensual studies point to the potential health benefits of implementing a moderate CR in the long-term on healthy young and middle-aged individuals, being a safe, well-tolerated approach, with no significant difference on reported adverse events [15]. Nonetheless, despite observational epidemiological studies revealed a reduction of mortality between 20–34% with CR, there is still insufficient evidence to support long-term nutritionally balanced CR as a safe strategy to increase lifespan [6]. Limitations of the analyzed studies include the impossibility to make inferences about BMI, monitoring of bone mass, possible anemia, and other deficiencies. An observational study along 15 years of CR showed long-term advantages, but there were no data to evaluate the adverse effects in the long run [6]. More and larger studies are needed to assess secondary events, malnutrition and reproductive system problems, as reported by Romashkan et al. [15], and also, the effects of the subsequent long-term metabolic rate reduction. Other biochemical/biological parameters must also be assessed, as well as the impact of CR diets on the gut microbiome. Despite short-term studies showed no differences in its composition [16], other studies demonstrate that ketosis modulates phylum-level microbiome, with a relative reduction of Firmicutes and Actinobacteria, and a slight growth of Bacteroides [44,47]. The clinical and physiological impact of this modulation of the microbioma must be studied, in special the impact of the pro/anti-inflammatory T Cells relationship.

The lack of studies on healthy non-obese individuals is a key limitation in this research area. Most studies are designed for obese or diabetic individuals, in which dietary approaches have a secondary prevention and treatment goal. Nonetheless, we consider primary prevention to be pivotal, and so we must understand how different dietary approaches reduce the long-term incidence of diseases.

The differences in the protocols found in the literature reinforce the need of further CR studies with different comparative strategies of ICR, in particular IF. It would be important to homogenize ICR protocols and IF typologies to better study their effects. This research would also help understanding how different protocols may be more beneficial for different individuals, contributing for the implementation of personalized diets, adapted to age, specific diseases, and other features. Other CR strategies are emerging and becoming increasingly popular, with extensively reported cases, yet some still with much empiricism. There is a need to better study these approaches and compare them with each other. We call attention to the results of VLCKD [25,70,71] and FMD [48,49,72,73]. A recent randomized study with FMD in neoadjuvant cancer chemotherapy, found very positive benefits as 90–100% loss of tumor cells, and prevention of DNA lesion on T cells [76]. This important evidence of FMD in cancer therapy must be explored in future trials. Additionally, as a suggestion to the scientific community, we propose trials to support the potential benefits of the exogenous use of ketones for rapid induction of a ketogenic environment and an appetite control strategy that allows adherence [14].

The management of metabolic and cardiovascular problems is well associated with regimens that treat IR, and so there is an urgent need for more robust studies that demonstrate whether the drastic restriction of sugars and tendency to induce ketogenesis is a safe way to promote rapid improvements in metabolic parameters in a more regular basis. As a common idea, ketosis is not recommended for seniors over 65 years old, and this raises a problem as this is the age group where cardiometabolic diseases are more prevalent. Consequently, there is still a need to study and define a target for calorie and carbohydrate restriction for this age group. Studies in different age groups with special emphasis on young teenagers and even in childhood are also needed, as there is an increasing prevalence of obesity and diabetes in these age groups and evidence is lacking.

Concluding, with the data from recent studies about metabolic regulation with CR dietary strategies (from the most classic low-calorie diet to the emerging low-carb ketogenic diet approaches), we are convinced that the paradigm that has guided dietary prescriptions and the work of physicians and scientists in the last decades (based on the food pyramid, 50–60% carbohydrates, and lipid restriction) for the prevention of chronic cardiovascular disease, dyslipidemia and diabetes, will have to change and adapt to the newest evidence. Changes in dietary paradigms have previously happened, such as with the false idea that sardines and eggs caused dyslipidemia, and well-designed robust studies are already challenging the current dietary paradigm. As evidence grows, we believe official guidelines will tend to dramatically reduce the percentages of carbohydrates, especially those derived from grains in the form of refined flours with a high glycemic index, and increase the percentage of unprocessed fat, animal or vegetable, preferably from sources of omega 3, 6 and 9 with appropriate proportions, and a normoproteic ratio as stipulated (0.8–1 gr/kg of weight). Similar proportions were evaluated with great results in the SKMD, based on fish, egg, poultry, and legumes as sources of protein, olive oil as the main source of fat and vegetables as a source of carbohydrates.

## Figures and Tables

**Figure 1 nutrients-13-01302-f001:**
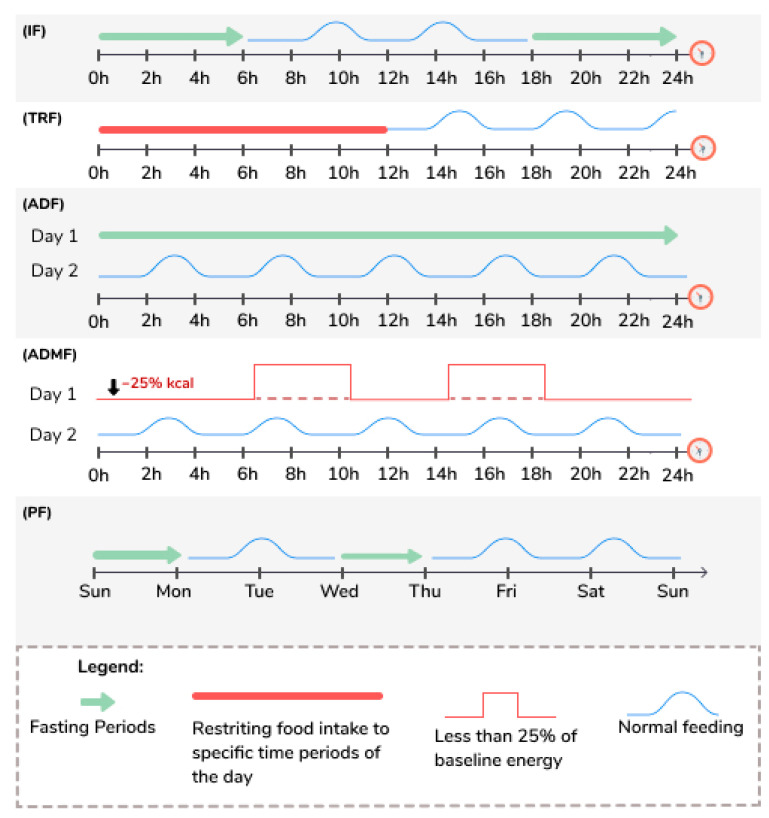
Illustration of the terms used to describe different types of fasting. ADF: Alternate Day Fasting; IF: Intermittent Fasting; ADMF: Alternate Day Modified Fasting; PF: Periodic Fasting; TRF: Time Restricted Feeding.

**Figure 2 nutrients-13-01302-f002:**
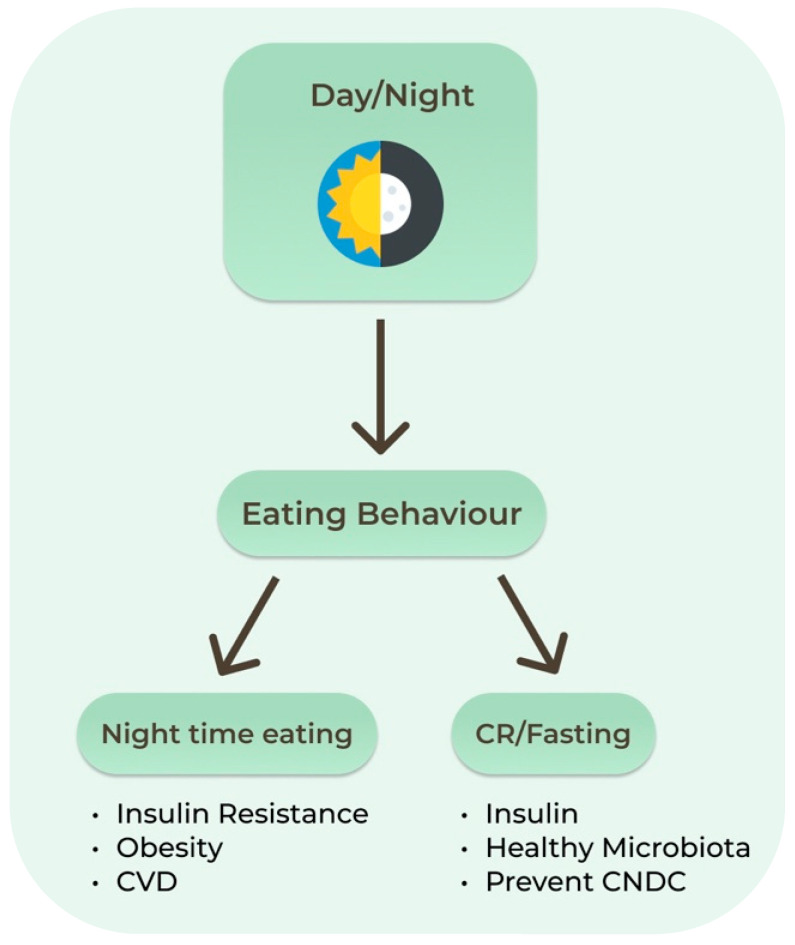
Circadian clock and effects of eating behaviors. CNCD: Chronic Non-communicable Diseases. CR: Calorie Restriction; CVD: Cardiovascular Diseases.

**Figure 3 nutrients-13-01302-f003:**
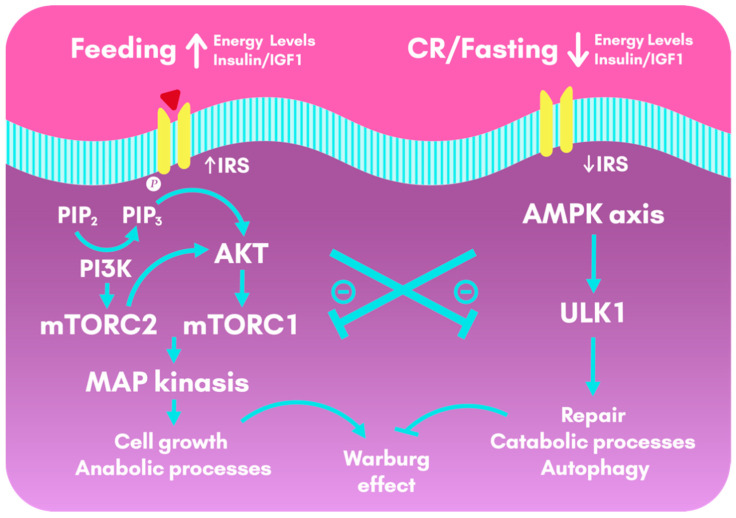
Cellular metabolic pathways activated during feeding and calorie restriction and fasting. During feeding, insulin and IGF1 lead to the activation of the insulin-pAKT-mTOR pathway, that drives downstream gene activities that promote anabolic processes, cell growth and cell survival. This pathway, although crucial for the organism, also favor the Warburg effect (aerobic glycolysis in cancer cells). In contrast, CR or a few hours of fasting activate AMPK, which triggers repair and catabolic processes, as well as autophagy. CR and fasting have an anti-Warburg effect by counteracting the insulin and IGF1 pathways. AKT: Protein Kinase B; AMPK: AMP-activated protein Kinase; CR: Calorie Restriction; IGF1: Insulin-like Growth Factor 1; IRS: Insulin Receptor Substrates; MAP: Mitogen-Activated Protein; mTOR: mechanistic Target OF Rapamycin; PI3K: Phosphoinositide 3-Kinase; PIP2: Phosphatidylinositol biphosphate; PIP3: Phosphatidylinositol triphosphate; ULK1: Serine/threonine-protein kinase 1.

**Table 1 nutrients-13-01302-t001:** Blood levels during a normal diet, ketogenic diet and diabetic ketoacidosis [26].

Blood Levels	Normal Diet	Ketogenic Diet	Diabetic Ketoacidosis
Glucose (mg/dL)	80–120	65–80	>300
Insulin (U/L)	6–23	6.6–9.4	≈0
Ketone bodies (mmol/L)	0.1	7/8	>25
pH	7.4	7.4	<7.3

**Table 2 nutrients-13-01302-t002:** Definition of the terms used to describe different types of fasting. Adapted from Anton et al. [51].

Intermittent Fasting (IF) [52,53,54,55,56,57,58,59,60,61]	This eating pattern involves fasting for varying periods of time, typically for 12 h or longer.
Time Restricted Feeding (TRF) [53,54,55,56,57,61]	This eating pattern involves restricting food intake to specific time periods of the day, typically between 8 to 12 h each day.
Alternate Day Fasting (ADF) [50,55,56,57,60,61,62]	This eating pattern involves consuming no calories on fasting days and alternating fasting days with a day of unrestricted food intake or “feast” day.
Alternate Day Modified Fasting (ADMF) [63,64]	This eating pattern involves consuming less than 25% of baseline energy needs on “fasting” days, alternated with a day of unrestricted food intake or “feast” day.
Periodic Fasting (PF) [52,53,54,55,56,57,59]	This eating pattern consists of fasting only 1 or 2 days/week and consuming food ad libitum on 5 to 6 days per week.

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
