# Peer review of "Effects of Calorie Restriction on Health Span and Insulin Resistance: Classic Calorie Restriction Diet vs. Ketosis-Inducing Diet"

_nutrients, 2021, doi:10.3390/nu13041302_

Round 1
Reviewer 1 Report
Nutrients:
Comparative effects of calorie restriction on health span and insulin resistance: classic calorie restriction diet us ketosis-inducing diet
Ana Napoleao, Livia Fernamdes, Ctia Miranda, Ana Paula Marum
This review article introduces not only the outline about the trial data of various CR mainly in human from the recent randomized articles and the benefits and problems containing mammalian experimental data. Particularly, recent many reports about various CR trials using volunteer deserve an attention for many researchers, physicians and ordinary people. However, there are a few constitutive problems (containing subtitles..) in the paragraphs and refinements. The details are mentioned in the following.
Major comments:
1) Would authors need the SECTION 2 (Methodology) and 3 (Results and Discussion) in this reviews? If so, this review article is probably not a review but the paper classified CR, and also the title should be revised (e.g., Classification of CR …in the recent reports). Especially, is SECTION 2 (Methodology) necessary in this review? Refer to the following, Minor comment of 2).
Minor comments:
1) P2L49: “Type 2 Diabetes (T2DM)” is correctly “Type 2 Diabetes Mellitus (T2DM)”?
2) P2L69: The authors mention about the criteria of selected reference. However, the criteria should be not ‘the authors’ interesting’ rather the randomized selection.
3) P2L73: The subtitle “3.1. Calorie Restriction” is suitable for example rather “3.1. Definition of Calorie Restriction”?
4) P2L90: The subtitle “3.1.1. Calorie Restriction, lifespan and health span” is suitable for example rather “3.1.1. Calorie Restriction, effects on lifespan and health span” or “3.1.1. Mechanisms of Calorie Restriction”?
5) P3L102: “chronic non-communicable diseases” to “CNCD”?
6) P4L159: “3.1.2. Calorie Restriction and insulin” is suitable for example rather “3.1.2. Calorie Restriction and effect of insulin” or “..and effect on insulin sensitivity”?
7) P5L241: “3.1.3. Calorie Restriction, insulin and exercise” is suitable for example rather “3.1.3. Calorie Restriction, interaction of insulin and exercise”?
8) P5L204 and L246: In these glucose concentration, different units of measure are used (mmol/L or mg/dL). Therefore, almost all readers may be confused the wide range of numerical value.
9) P6L269: “3.2. Calorie Restriction and Ketosis-inducing diet” is suitable for example rather “3.2. Definition of Ketosis-inducing Diet in Calorie Restriction”?
10) P7L302: “3.2.1. Ketogenic Diet” is suitable for example rather “3.2.1. Difference between Ketogenic Diet and Ketosis-inducing Diet”?
11) P7L336: “HgA1c” to “HbA1c”.
12) P8L350: “hydroxybutyrate” correctly to “b-hydroxybutyrate”.
13) P8L356: The subtitle “3.2.2. Intermittent Fasting” is contained indeed in “3.2. Calorie Restriction and Ketosisi-inducing diet”? Perhaps, another subtitle (e.g., “3.3. Definition and Mechanisms of Intermittent Fasting”) may be suitable.
14) P8L376-378: Maybe, the authors should show the schema (such as Figures) to explain the molecular mechanisms containing the insulin-pAKT-mTOR pathway and AMPK in cells based on studies using model organisms.
15) P9L385-386: “type 2 insulin-dependent diabetes” maybe to “insulin-dependent T2DM”?
16) P12L555: “chronic non-communicable diseases” to “CNCD”?
Author Response
As a representative of the group of authors of the review article “Comparative effects of calorie restriction on health span and insulin resistance: classic calorie restriction diet vs keto-sis-inducing diet”, we thank you and appreciated the careful, and detailed revision. Your suggestions and comments were received as a challenge which provide an opportunity to optimize our work for Nutrients. We consider all points relevant to improve the article.
We are grateful for the score considered in the criteria: considering a well-organized and comprehensively work with correct English and significant contribution to the field, scientifically sound appropriate and adequate references.
We agree with all the appointed constitutive problems and we expect it to be resolved: we eliminate the “Methodology” and the “Results”. We respect the format of a review article despite the confusion of having followed an incorrect template.
After checking the final result of this extensive revision, proposed by the reviewers of our article, we considered adapting the title to the current content. We deleted the word "Comparative" as this review goes beyond a comparison of the effects of two nutritional strategies, but it contains an extensive review of the effects of the classic caloric restriction versus ketosis-inducing diets on Health span and insulin metabolism. So instead of “Comparative Eeffects of calorie restriction on health span and insulin resistance: classic calorie restriction diet vs keto-sis-inducing diet” we suggested “Effects of calorie restriction on health span and insulin resistance: classic calorie restriction diet vs ketosis-inducing diet
We found all the minor comments pertinent and well done. We try to correct them all, and we expect to have solved them, agreeing with what was proposed to us.
1) P2L49: “Type 2 Diabetes (T2DM)” is correctly “Type 2 Diabetes Mellitus (T2DM)”?
Error corrected. See L 52
2) P2L69: The authors mention about the criteria of selected reference. However, the criteria should be not ‘the authors’ interesting’ rather the randomized selection.
Correct. See P3L47-49. We regret that the description induced that it had not been random and randomized.
3) P2L73: The subtitle “3.1. Calorie Restriction” is suitable for example rather “3.1. Definition of Calorie Restriction”?
We agree and corrected for “2. Calorie Restriction, definition and effects” L60
4) P2L90: The subtitle “3.1.1. Calorie Restriction, lifespan and health span” is suitable for example rather “3.1.1. Calorie Restriction, effects on lifespan and health span” or “3.1.1. Mechanisms of Calorie Restriction”?
We agree and corrected for “2.1. Calorie Restriction, effects on lifespan and health span”. See L77
5) P3L102: “chronic non-communicable diseases” to “CNCD”?
Correct. See L 89
6) P4L159: “3.1.2. Calorie Restriction and insulin” is suitable for example rather “3.1.2. Calorie Restriction and effect of insulin” or “..and effect on insulin sensitivity”?
We agree and corrected for “2.2 Calorie Restriction and the effects on insulin sensitivity”. See L146
7) P5L241: “3.1.3. Calorie Restriction, insulin and exercise” is suitable for example rather “3.1.3. Calorie Restriction, interaction of insulin and exercise”?
We agree and corrected for “2.3. Calorie Restriction, interactions between insulin and exercise”. See L 228
8) P5L204 and L246: In these glucose concentration, different units of measure are used (mmol/L or mg/dL). Therefore, almost all readers may be confused the wide range of numerical value.
Yes, glucose concentration had mixed units, so we choose to convert the units to mg/dl. See L191, 315
9) P6L269: “3.2. Calorie Restriction and Ketosis-inducing diet” is suitable for example rather “3.2. Definition of Ketosis-inducing Diet in Calorie Restriction”?
We agree and changed the title for “3. Ketosis-inducing diets, definition, types and effects”. See 256
10) P7L302: “3.2.1. Ketogenic Diet” is suitable for example rather “3.2.1. Difference between Ketogenic Diet and Ketosis-inducing Diet”?
We decided to change the title to “3.1Ketogenic Diet, definition and effects”, making the title more specific. We believe this title is more suitable as the chapter 3.1 includes more information, and it was also completed during this revision. See L289
11) P7L336: “HgA1c” to “HbA1c”.
Was corrected. See L 401
12) P8L350: “hydroxybutyrate” correctly to “b-hydroxybutyrate”.
We appreciate the correction of this detail. See L 337
13) P8L356: The subtitle “3.2.2. Intermittent Fasting” is contained indeed in “3.2. Calorie Restriction and Ketosisi-inducing diet”? Perhaps, another subtitle (e.g., “3.3. Definition and Mechanisms of Intermittent Fasting”) may be suitable.
3.2. Intermittent Fasting, definition and effects, is included in the chapter “3. Ketosis-inducing diets, definition, types and effects”, as this chapter includes all the diets that induce ketosis, as explained in the beginning of the chapter: “Ketosis-inducing diets represent a group of diets that induce the production of ketones, comprising: diets with low carbohydrate content, normoproteic content and high-fat content, such as the Classic Ketogenic Diet (CKD) and the Fasting mimicking diet (FMD); and diets in which the individual restricts the eating period, having intermittent ketosis, such as Intermittent fasting (IF)”. See L355-415
14) P8L376-378: Maybe, the authors should show the schema (such as Figures) to explain the molecular mechanisms containing the insulin-pAKT-mTOR pathway and AMPK in cells based on studies using model organisms.
We appreciate the suggestions, and we had the opportunity to prepare a Figure with the molecular mechanisms proposed. Figure for relation between Eating behaviour in circadian cycle and CNCD was included, and a schematic figure for fasting type diets.
See Figure 1. Illustration of the terms used to describe different types of fasting L374-375
Figure 2. Circadian clock and effects of eating behaviors L396-397
Figure 3. Cellular metabolic pathways activated during feeding and calorie restriction and fasting L588-589
15) P9L385-386: “type 2 insulin-dependent diabetes” maybe to “insulin-dependent T2DM”?
Correct. See L 398
16) P12L555: “chronic non-communicable diseases” to “CNCD”?
Correct. See L 715

Reviewer 2 Report
Ana Napoleao and collaborators provided a comprehensive review on a hot topic, that is the effect of calorie restriction on health span and as a non pharmacological intervention to prevent non communicable diseases. Various types of calorie restriction schemes are discussed including the ketogenic diet. The review is well structured and articulated. I have thus only minor suggestions. Since in the title there is a specific mention to the comparison between classic calorie restriction diet versus ketogenic diet, in my opinion, the section 3.2.1 on ketogenic diet should be expanded discussing also the beneficial effects it exerts in cancer (targeting the Warburg effect) and neurodegenerative diseases. Some recent and important references on the topic should also be included. Moreover, a mention on the influence of ketogenic diet on gut microbiota may also be added.
The same applies to the section 3.2.4 on fasting mimicking diet where beneficial effects have been reported in recent important papers (appeared in 2020) in cancer and cardiovascular diseases which need to be added.
In general I would like to suggest the authors to highlight the concept that there is no 'one fits all' calorie restriction diet. Although calorie restriction is well documented to improve life span and represents a powerful tool to preserve health, different types of approach in quantity and quality of calories eaten should be used when dealing with treatment of specific diseases, such as metabolic diseases, cardiovascular diseases, and cancer, also in consideration of the age of patients.
Time restricted eating and circadian clock are other important aspects to consider in the management of metabolic diseases and may deserve further discussion in this review. This aspect is only briefly cited in section 3.2.2.
Lastly, one or two figures may help to focus the reader's attention on the key concepts implied in calorie restriction diets, e.g. different timing schemes, balanced diet versus high fat diet, insulin sensitivity and development of insulin resistance.
Author Response
As a representative of the group of authors of the review article “Comparative effects of calorie restriction on health span and insulin resistance: classic calorie restriction diet vs keto-sis-inducing diet”, we thank you and appreciated the careful, and detailed revision. Your suggestions and comments were received as a challenge which provide an opportunity to optimize our work for Nutrients. We consider all points relevant to improve the article.
We are grateful for the score considered in the criteria: considering a well-organized and comprehensively work with correct English and significant contribution to the field, scientifically sound appropriate and adequate references.
We found all the minor comments pertinent and well done. We try to correct them all, and we expect to have solved them, agreeing with what was proposed to us.
- After checking the final result of this extensive revision, proposed by the reviewers of our article, we considered adapting the title to the current content. We deleted the word "Comparative" as this review goes beyond a comparison of the effects of two nutritional strategies, but it contains an extensive review of the effects of the classic caloric restriction versus ketosis-inducing diets on Health span and insulin metabolism. So instead of “Comparative Eeffects of calorie restriction on health span and insulin resistance: classic calorie restriction diet vs keto-sis-inducing diet” we suggested “Effects of calorie restriction on health span and insulin resistance: classic calorie restriction diet vs ketosis-inducing diet.”
- A constitutive change in the article was made, by eliminating the "Methodology" chapter, with information about it in the introduction. Was deleted the title "Results" to adapted to the structure of the a article
- We agree with the suggestion to expand the discussion about the beneficial effects of Calorie restriction and Ketogenic diet on cancer (targeting the Warburg effect) and neurodegenerative diseases:
Was created new Chapters “4.Calorie Restriction and Ketosis-inducing diets in cancer “.
Therefore, we restructured the chapters and added the chapter that highlights calorie restriction, ketosis and cancer and introduced neurodegenerative diseases on Ketosis diet and the topic ketogenic diet on gut microbiota.
Introduced studies and the description relating Fasting mimicking diets with benefits in cancer
- The concept that there is no 'one fits all' calorie restriction diet was discussed in more detail in chapter 6. Conclusions. We hope this idea is clearer and more comprehensive now.
- To focus the readers, we strive to create figures that reflect the general message of the biochemical mechanisms of calorie restriction and its effects, different types of fasting diets, and the effects of nutritional disregard for the circadian rhythm and.
We have the expectation of valuing the article.
See:
Figure 1. Illustration of the terms used to describe different types of fasting
Figure 2. Circadian clock and effects of eating behaviours
Figure 3. Cellular metabolic pathways activated during feeding and calorie restriction and fasting

Round 2
Reviewer 1 Report
Recommendation: The authors have been submitted their response with a clear and relevant revision, therefore, I recommend to publish with revision in this journal.